# MetaEnzyme: Meta Pan-Enzyme Learning for Task-Adaptive Redesign

## ABSTRACT

Enzyme design plays a crucial role in both industrial production and biology. However, this field faces challenges due to the lack of comprehensive benchmarks and the complexity of enzyme design tasks, leading to a dearth of systematic research. Consequently, computational enzyme design is relatively overlooked within the broader protein domain and remains in its early stages. In this work, we address these challenges by introducing MetaEnzyme, a staged and unified enzyme design framework. We begin by employing a cross-modal structure-to-sequence transformation architecture, as the feature-driven starting point to obtain initial robust protein representation. Subsequently, we leverage domain adaptive techniques to generalize specific enzyme design tasks under low-resource conditions. MetaEnzyme focuses on three fundamental low-resource enzyme redesign tasks: functional design (FuncDesign), mutation design (MutDesign), and sequence generation design (SeqDesign). Through novel unified paradigm and enhanced representation capabilities, MetaEnzyme demonstrates adaptability to diverse enzyme design tasks, yielding outstanding results. Wet lab experiments further validate these findings, reinforcing the efficacy of the redesign process. *The code and data will be publicly released.*

## CCS CONCEPTS

• **Computing methodologies** → **Artificial intelligence**; • **Applied computing** → **Computational biology**; **Life and medical sciences**; *Bioinformatics*.

## KEYWORDS

Protein Design, Functional Prediction, Mutation Effect, Sequence Generation, Enzyme Engineering

## 1 INTRODUCTION

Enzymes, distinguished as specialized proteins, serve as biological catalysts, expediting chemical reactions. Their capacity to catalyze reactions ensures specificity and enables operation under mild conditions, thus playing a crucial role in various industries. [1, 34, 47, 49]. Positioned at the forefront of industrial production and the biological domain, enzyme design involves the deliberate creation of modified variants through functional design, commonly known as protein redesign [6, 21, 37], based on known structures or sequences.

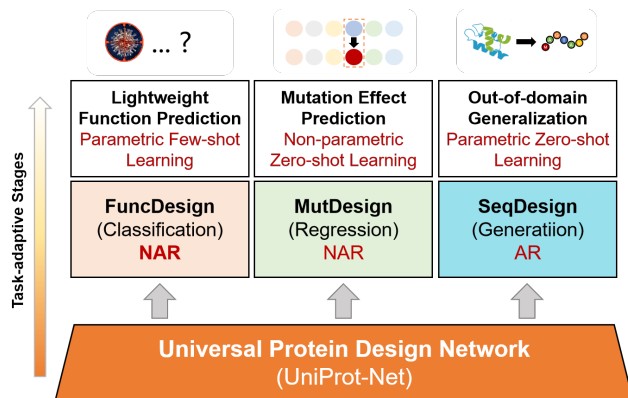

**Figure 1: The overall logical framework of MetaEnzyme and the successive phases for various enzyme functional tasks.**

Despite its critical role, computational enzyme design is still in its early stages within the broader protein field. The scarcity of comprehensive enzyme data, coupled with the diversity of enzyme tasks and models [30, 52], has resulted in a lack of systematic research and oversight in computational enzyme design. The inherent complexity of tasks and the vast diversity of data pose challenges for widespread adoption, contributing to relatively low attention in the enzyme field.

To address the challenges inherent in enzyme design, it is imperative to confront issues stemming from data scarcity and model generalization. Enzyme datasets often suffer from smaller scales compared to the broader protein domain due to specific functional categorizations, leading to inadequate model training. To mitigate this challenge, we propose leveraging pretrained universal protein models as intermediaries, replacing direct training of task-specific enzyme functional models. These universal models benefit from more extensive datasets in the general domain, resulting in stronger representation capabilities in both sequence and structural modalities. This could facilitates better domain adaptation to downstream tasks through transfer learning manner.

Furthermore, functional enzyme tasks exhibit complex diversities, necessitating different modalities and data requirements, which may require multi-modality pretrained model architectures as drivers. However, this approach is not the more efficient or unified solution. Given our concentration on essential low-resource enzyme redesign tasks—functional design (FuncDesign), mutation design (MutDesign), and sequence generation design (SeqDesign)—encompassing the primary tasks and paradigms of contemporary AI research in enzymes, we underscore the significance of incorporating structural modality. This emphasis stems from the recognized principle that structures dictate functions. Taking these considerations into account, we aim to pursue both robust representation and generalization while simplifying the enzyme design framework.

Through observing commonalities across all tasks, we propose a novel MetaEnzyme framework, as illustrated in Figure 1. MetaEnzyme consists of a foundational universal protein design network (UniProt-Net) and downstream enzyme redesign modules. In the first stage, UniProt-Net undergoes pretraining to acquire robust representation capabilities. We instantiate UniProt-Net with a cross-modal structure-to-sequence network, as illustrated in Figure 2(a), to bridge the gaps between various redesign tasks. To augment structural and contextual representations, we introduce geometry-equivariant modules, implicit energy-motivated data augmentation, and multi-modal fusion, significantly enhancing generalization capabilities for low-resource scenarios. In the second stage, all enzyme redesign modes are driven by the cross-modal UniProt-Net. Based on characteristics such as autoregressive (AR) vs. non-autoregressive (NAR), non-parametric vs. parametric, etc., we rationalize different tasks and adopt meta-learning and domain-adaptive techniques in low-cost and high-efficiency manners.

The main contributions are outlined as follows:

• The innovative unified enzyme design framework, MetaEnzyme, consolidates patterns across mainstream tasks, effectively leveraging commonalities for enhanced adaptability. The framework facilitates seamless transitions between universal protein design network and in-domain design evaluation tasks.

• The proposed geometry-enhanced module, combined with techniques such as data augmentation, multi-modal fusion, significantly enhances representation capabilities for low-resource settings, along with the extension of multiple enzyme tasks.

• A novel decoupled mutation scoring method is introduced for evaluating mutation effects, greatly improving efficiency.

• Additional wet lab experiments for computational validation.

## 2 UNIVERSAL PROTEIN DESIGN NETWORK

### 2.1 Problem Statement

As shown in Figure 2(a), UniProt-Net aims to to generate amino acid sequences conditioned on the protein backbones. UniProt-Net comprises a Geometric-invariant Structure Encoder (GeoStruc-Encoder), a Structure-Sequence Adapter (StrucSeq-Adapter), and a Self-attention Sequence Decoder (SaSeq-Decoder). The GeoStruc-Encoder and StrucSeq-Adapter collectively constitute the Encoder-Adapter Module (EnAd-Module), responsible for ensuring transformation equivariance. The StrucSeq-Adapter and SaSeq-Decoder collectively constitute the Context-Module incorporating fully self-attention layers.

Formally, given a protein backbone $X = \{X_1, X_2, \cdots, X_n\}$ with a length of $n$, where $X_i \in \mathbb{R}^{3 \times 3}$ represents the atomic coordinates of the i-th amino acid residue composed of N, $C_\alpha$, and C atoms. The corresponding generated protein sequence is denoted as $Y = \{Y_1, Y_2, \cdots, Y_n\} \in \mathbb{R}^n$. And $\hat{Y}$ denotes the native sequence $\hat{Y} = \{\hat{Y}_1, \hat{Y}_2, \cdots, \hat{Y}_n\} \in \mathbb{R}^n$. The function $p_{uniprot}$ represents the underlying UniProt-Net.

### 2.2 The Underlying Structure-to-Sequence Network

#### 2.2.1 Geometry-enhanced Structural Encoder. **Definition 1**. *For any transformation $\mathcal{T} \in E(3)$, a geometric network $\varphi$ is E(3)-equivariant if $\varphi(\mathcal{T} \cdot X) = \mathcal{T} \cdot \varphi(X)$, and $\varphi$ is E(3)-invariant if $\varphi(\mathcal{T} \cdot X) = \varphi(X)$.*

The core component of the GeoStruc-Encoder is the lightweight GVP module [19], in which the vanilla GVP layers are overall rotation-invariant for rigid bodies since GVP outputs a rotation-equivariant vector feature $v$ through an equivariant function $f$, and a rotation-invariant scalar feature $s$ through an invariant function $g$ for each amino acid, thus for any arbitrary rotation $\mathcal{R}$: $\mathcal{R}f(v) = f(\mathcal{R}v), g(s) = g(\mathcal{R}s)$, as shown in Figure 2(b).

Then to make the entire EnAd-Module rotation-invariant, a local reference frame $v_l$ [15] is further introduced to fuse with the rotating vector feature $v'$, resulting in enhanced rotation-invariant features, which is rotation-invariant for any rotation $\mathcal{R}$: $f(v' \oplus v_l) = f(\mathcal{R}(v' \oplus v_l))$. Additionally, the input features are translation-invariant[20], making the overall EnAd-Module also translation-invariant. Therefore, the concatenated features are invariant to translations and rotations on the input coordinates. Thus for any rotation/translation $\mathcal{T}$ of the input, the output of the entire EnAd-Module $p_{enad}$ can be invariant: $p_{enad}(v, s) = p_{enad}(\mathcal{T}(v, s))$. To obtain a better structural representation, we initialize the EnAd-Module parameters as [15] which is trained on millions of proteins.

#### 2.2.2 Energy-motivated Data Augmentation. Guided by biochemistry principles, atomic interactions are regulated by forces and energy. This insight inspires the incorporation of energy and force concepts into data augmentation[17, 18]. To achieve this, we introduce an energy-driven Riemann-Gaussian geometric technique, illustrated in Figure 2(c), which ensures that structural variations maintain the energy dynamics of proteins, preserving pairing relationships and geometric characteristics. In essence, the augmented structures guarantee structural invariance for the spatial distribution of energy or force. In a nutshell, we obtain a noisy sample $X'$ from $X$ according to a certain conditional distribution, i.e., $X' \sim p_{noise}(X'|X)$. Unlike conventional denoising methods applied to images or other Euclidean data, the introduced noise in our 3D geometry is tailored to be geometry-aware rather than conformation-aware, i.e., $p(X'|X)$ should possess doubly E(3)-invariant:

$$p(\mathcal{T}_1 \cdot X'|\mathcal{T}_2 \cdot X) = p(X'|X), \forall \mathcal{T}_1, \mathcal{T}_2 \in E(3). \quad (1)$$

This is consistent with the observation that the behavior of proteins with the same geometry should be independent of different conformations. A conventional choice of $p_{noise}(X'|X)$ is utilizing the standard Gaussian with noise scale $\sigma$ as $p_{noise}(X'|X) = \mathcal{N}(X, \sigma^2 I)$. But this naive form fails to meet the doubly E(3)-invariant property in Eq. 1. Specifically, considering the derived force target $\nabla_{X'} \log p(X'|X) = -\frac{X'-X}{\sigma^2}$ where $X' = \mathcal{R} \cdot X$ s.t. rotation $\mathcal{R} \neq I$, then $\nabla_{X'} \log p(X'|X) = -\frac{1}{\sigma^2}(\mathcal{R} - I)X \neq 0$, which imply that the force between $X' = \mathcal{R} \cdot X$ and $X$ is not equal although the same geometry is shared. Hence, to devise the form with the symmetry in Eq. 1, we instead resort to Riemann-Gaussian [7] as:

$$p_{noise}(X'|X) = \text{Rie}_\sigma(X'|X) := \frac{1}{\zeta(\sigma)} \exp(-\frac{\delta^2(X', X)}{4\sigma^2}), \quad (2)$$

where $\zeta(\sigma)$ is the normalization term, and $\delta$ is the metric that calculates the difference between $X'$ and $X$. Riemann-Gaussian is a generalization version of typical Gaussian, by choosing various distances $\delta$ beyond the Euclidean metric. To pursue the constraint in Eq. 1:

$$\delta(X', X) = ||X_r'^T X_r' - X_r^T X_r||_2, \quad (3)$$

**Figure 2: (a) The underlying UniProt-Net instantiation is based on a structure-to-sequence framework including a Geometry-invariant Structural Encoder (GeoStruc-Encoder), a Structure-Sequence Adapter (StrucSeq-Adapter), and a Self-attention Sequence Decoder (SaSeq-Decoder). (b) Elaborate feature modules within the GeoStruc-Encoder. (c) Structural data augmentation employing Riemann-Gaussian noise. (d) Initialization of the Context-Module (StrucSeq-Adapter & SaSeq-Decoder) in a self-supervised manner.**

where $X_r = X - \mu(X)$ shifts $X$ to zero mean ($\mu(X)$ is the mean of $X$'s columns). The same transformation applies to $X_r'$. The distance function $\delta$ adheres to the doubly E(3)-invariance as in Eq. 1. Notably, $\delta$ is permutation-invariant concerning the order of the columns in $X'$ and $X$.

*2.2.3 Context Module Initialization.* The initialization of the Context-Module, as shown in Figure 2(d), aims to incorporate prior language knowledge into the modules[51]. The entire Context-Module functions as an encoder-decoder Transformer[45], denoted as $p_{context}$ with linear output layers. To initialize the Context-Module exclusively based on sequence data from the training set, we employ a sequence-to-sequence recovery task using an autoencoder (AE) mode. This allows the Context-Module to acquire contextual semantic knowledge, utilizing cross-entropy (CE) loss for learning. Formally, given an input protein sequence for the encoder $S_{in} = \{s_1, s_2, \cdots, s_n\}$ with $n$ amino acids, and the reference native sequence as $S_{native} = S_{in} = \{s_1, s_2, \cdots, s_n\}$, the AE-based objective is to generate a sequence $S_r$ to recover $S_{native}$ as accurately as possible:

$$\mathcal{L}_{AE} = CE\big(p_{context}(\text{logits}_{S_r}|S_{in}), S_{native}\big). \tag{4}$$

## 2.3 Training Pipeline in Initial Phase

We initialize the parameters of UniProt-Net based on prior knowledge. To enhance the dataset, we incorporate Riemann-Gaussian data augmentation, combining the original protein dataset $D_{ori}$ with the augmented dataset denoted as $D_{rie}$, resulting in a new dataset $D_{aug} = D_{ori} \cup D_{rie}$. In the processing flow, the protein backbone $X \sim D_{aug}$ is initially input into the EnAd-Module, producing geometric context features. These features are then fed into the

decoder to generate the sequence distribution logits$_Y$ and derive the generated sequence $Y$ as:

$$Y = \text{argmax}\big(p_{uniprot}(\text{logits}_Y|X)\big), X \sim D_{aug}, \tag{5}$$

and cross-entropy loss is used for training:

$$\mathcal{L}_{\text{primary}} = CE(p_{uniprot}(\text{logits}_Y|X), \hat{Y}). \tag{6}$$

# 3 DOMAIN ADAPTION FOR LOW-RESOURCE ENZYME DESIGN TASKS

---

**Algorithm 1** Task-specific Meta Enzyme Learning

---

**Require:** $T$: task distributions; $T_i$: individual enzyme task s.t. $i \in [0, N)$; $L_{out}$ and $L_{in}$: Update steps for outer and inner loop; $bt$ instances in each batch of tasks; $\ell, \ell'$: learning rates; Randomly initialize $\theta$;

**repeat**
  **if** $L_{in} > 0$ **then**
    Sample batch of tasks $T_i \sim T$.
    **for** all $i \in [0, N)$ **do**
      Evaluate $\nabla_\theta \mathcal{L}_{T_i}(f_\theta)$ with respect to $bt$ examples;
      Compute adapted parameters with gradient descent $\theta_i' = \theta - \ell \cdot \nabla_\theta \mathcal{L}_{T_i}(f_\theta)$;
    **end for**
    Update $\theta \leftarrow \theta - \ell' \cdot \nabla_\theta \sum_{T_i \sim T} \mathcal{L}_{T_i}(f_{\theta_i'})$;
    $L_{in}$ -= 1;
  **end if**
  $L_{out}$ -= 1;
**until** $L_{out} == 0$;

---

## 3.1 Few-shot Learning for Functional Design (FuncDesign)

We introduce a task-specific meta-learning framework [5, 11]for the task distribution $T$, outlined in Algorithm 1. The UniProt-Net here is regarded as a function $f_\theta$ with an initial pretrained parameter $\theta_{meta}$ involving linear classification layers. The primary objective of the meta-learner is to acquire updated parameters of $\theta_{meta}$ containing high-order meta-information, facilitating swift adaptation to a new task drawn from $T$. Upon adapting the meta-learner to a new task $T_i \sim T$, the $\theta_{meta}$ undergoes updates, transforming into the task-specific parameter $\theta_i$ through a few gradient descent iterations. These parameter updates, executed based on the support set $S_i$ for $T_i$, are referred to as inner loops. The number of inner loops is denoted as $L_{in}$. And loss function $\mathcal{J}_{meta}$ is utilized in the inner-loop updates as:

$$\theta_i^{(in)} = \theta_{meta} - \sum_{t=0}^{L_{in}-1} (\ell \cdot \nabla_{\theta_i^{(t)}} \mathcal{L}_{S_i}^{(t)}) \quad s.t.$$

$$\mathcal{L}_{S_i}^{(t)} = \mathbb{E}_{X_i, \hat{Y}_i, \hat{c} \sim S_i} \left( \mathcal{J}_{meta}(f_{\theta_i^{(t)}}(c_i|\hat{Y}_i; X_i), \hat{c}_i) \right), \quad (7)$$

where $\theta_i^{(t=0)}$ equals to $\theta_{meta}$, and $\theta_i^{(t)}$ denotes the fold-specific parameters after $t$ inner loops. $\ell$ is the learning rate. $X_i, \hat{Y}_i$ represents the native structure-sequence pairs, with $\hat{c}_i$ as labels in $S_i$. The optimization of $\theta_{meta}$ occurs on the query set $Q_i \sim Q$ across tasks $T$ during outer loops. Consequently, the update for $\theta_{meta}$ at one step as:

$$\theta_{meta} \leftarrow \theta_{meta} - \sum_{Q_i}^{Q} (\ell' \cdot \nabla_{\theta_{meta}} \mathcal{L}_{Q_i}) \quad s.t.$$

$$\mathcal{L}_{Q_i} = \mathbb{E}_{X_i', \hat{Y}_i', \hat{c}' \sim Q_i} \left( \mathcal{J}_{meta}(f_{\theta_i^{(in)}}(c_i'|\hat{Y}_i'; X_i'), \hat{c}_i') \right) \quad (8)$$

where $\ell'$ is the learning rate of the outer loop. $X_i', \hat{Y}_i'$ represent protein pairs with $\hat{c}_i'$ labels in $Q_i$, similarly. The meta-objective function $\mathcal{J}_{meta}$ varies across functional tasks. For instance, in binary categorization, it can be expressed as:

$$\mathcal{J}_{meta} = -y(1-p)^\gamma \log(p) - (1-y)p^\gamma \log(1-p)$$

$$= \begin{cases} -(1-p)^\gamma \log(p), & y = 1 \\ -p^\gamma \log(1-p), & y = 0 \end{cases} \quad (9)$$

where $\gamma > 0$ denotes an adjustable factor. While in a multi-classification task, it can be expressed as :

$$\mathcal{J}_{meta} = -\sum_{i=1}^{K} y_i \log(p_i), \quad (10)$$

where $K$ is the number of classification categories.

## 3.2 Zero-shot Learning for Mutant Effect Prediction (MutDesign)

Mutation effect prediction is a non-parametric zero-shot learning in the NAR manner. Given the unique architecture of our MetaEnzyme, we introduce a novel mutation effect scoring method that decouples mutation scoring approaches, as illustrated in Figure 3. Formally, considering a wild-type enzyme structure $X_{wild} \in \mathbb{R}^{n \times 3 \times 3}$ with its

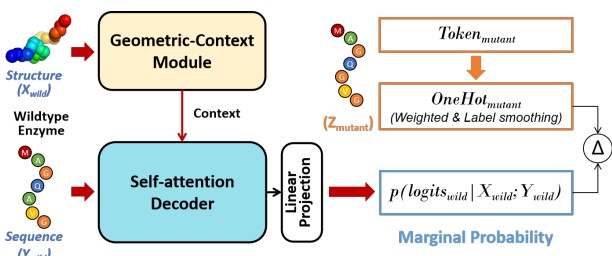

**Figure 3: The proposed decoupled mutational effect scoring.**

corresponding wild-type sequence $Y_{wild} \in \mathbb{R}^n$, the non-parametric mutation effect prediction task aims to query the mutation-related ranking for the mutation sequence $Z_{mut} \in \mathbb{R}^n$, which is scored based on the marginal probabilities $logits_{wild}$ and the reference distribution $logits_{mut}$ as:

$$\rho(Y_{wild}||Z_{mut}) = \delta_{mut}\big(logits_{wild}, LS(logits_{mut})\big),$$

$$s.t. \quad logits_{wild} = p_{uniprot}(X_{wild}; Y_{wild}) \in \mathbb{R}^{n \times M},$$

$$= \{y_0, y_1, \cdots, y_n\} \quad (11)$$

$$logits_{mut} = OneHot(Z_{mut}) \in \mathbb{R}^{n \times M},$$

$$= \{z_0, z_1, \cdots, z_n\}$$

where $\delta_{mut}$ is a distance measure between distributions of the wild-type and mutant. $M=20$ corresponds to 20 commonly used amino acid types. $OneHot$ is one-hot encoding, with $z_{ik} \in \{0, 1\}$ s.t. $z_{ik} \sim z_i, 0 \le k < M$. $LS$ denotes the label smoothing function with a constant $\epsilon = 0.1$ as:

$$LS(logits_{mut}) = logits_{mut} \times (1 - \epsilon) +$$

$$(1 - logits_{mut}) \times \frac{\epsilon}{M-1} = \{z_0', z_1', \cdots, z_n'\}. \quad (12)$$

Instantiating $\delta_{mut}$ with a weighted cross-entropy function:

$$\rho(Y_{wild}||Z_{mut}) = -W \cdot \sum_{i=1}^{n} z_i' \log(y_i), \quad (13)$$

where $W = \{w_1, w_2, \cdots, w_n\} \in \mathbb{R}^{n \times 1}$ is a weight matrix:

$$w_i = \begin{cases} \frac{1-\alpha}{\sum_{j=1}^{n} MutSet_j}, & MutSet_i = 1 \\ \frac{\alpha}{\sum_{j=1}^{n}(1-MutSet_j)}, & MutSet_i = 0 \end{cases} \quad (14)$$

where $MutSet \leftarrow (Y_{wild} == Z_{mut}) \in \mathbb{R}^n, \forall MutSet_i \in \{0, 1\}$, and $MutSet$ implies all mutant positions. $\alpha = 0.5$.

From Eq. 11, we observe that: 1) the computation of wild-type and mutant proteins is independent; 2) deep network flow (such as UniProt-Net) predicts only the wild-type proteins, requiring a single inference due to the insight that mutant sequences stem from unique wild-type proteins through evolution or modification; 3) large-scale mutant sequences do not necessitate a network model, and they can be converted to a distribution representation based on one-hot encoding simply. This decoupling method provides a significant advantage in terms of speed, particularly for sequence datasets exceeding a million entries and for scenarios involving multi-site/higher-order mutations. Additionally, as the computation process relies entirely on the pre-trained parameters of UniProt-Net and makes no distribution assumptions about the query mutant enzymes, it is non-parametric. Section 4.3.2 presents detailed analysis.

## 3.3 Autoregressive Generalization for Sequence Generation (SeqDesign)

For the conditional sequence generation, we shift the entire framework to an AR mode. Specifically, when provided with a candidate structure backbone $X_c$, the objective is to generate an unknown protein sequence $Y_{sample}$ as:

$$Y_{sample} = \prod_{i=0}^{n} p_{uniprot}(y_i | y_{i-1}, \cdots, y_0; X_c). \quad (15)$$

This density is represented using a Vanilla Transformer decoding paradigm. AR decoding typically conducts SeqDesign tasks for general proteins and functional enzymes.

## 4 IN SILICO EXPERIMENTS

### 4.1 Setups

**Training Set**. **CATH** dataset [16] is divided into training, validation, and testing sets, containing 18,204, 608, and 1,120 structure-sequence pairs, respectively. The training set is utilized to train the UniProt-Net in the pretraining stage.

**Evaluation Sets**. **FuncDB$_{bi}$**: Approximately 700 enzymes with experimentally determined structure-sequence pairs across 10 folds from RCSB are collected. Each fold is crafted with an equal number of enzymes and balanced with non-enzyme negative samples. This dataset is used for enzyme function prediction and design. **FuncDB$_{multi}$** is a subset of $FuncDB_{bi}$ dataset, comprising solely positive samples, for multi-class functional prediction tasks—precisely predicting the fold level to which an enzyme belongs. **MutDB$_{ProtGym}$** [36] is intricately designed for protein fitness, encompassing an expansive collection of over 217 deep mutational scanning assays and millions of mutant sequences. **SeqDesignDB$_{Pet}$** comprises 178 proteins identified for the capability to degrade plastics [12]. **SeqDesignDB$_{Hybrid}$** is the alias of $FuncDB_{multi}$ for differentiation. **SeqDesignDB$_{ALL}$** is extracted from the testing set of CATH. **SeqDesignDB$_{Ts50}$** and **SeqDesignDB$_{Ts500}$** correspond to Ts50 and Ts500 [26] for validating generalization.

**Implementation.** The AdamW optimizer with a batch size of 5 and a learning rate of *1e-3* to train the UniProt-Net. The GeoStruc-Encoder consists of 4 layers with a dropout of 0.1. The node hidden dimensions of scalars and vectors are 1024 and 256. The adapter and decoder have 8 multi-head self-attention layers, with an embedding dimension of 512 and an attention dropout of 0.1. For meta-learning configures, the number of inner loops is set to 5, and the batch sizes of the support set and query set each are both 10. The updated learning rates $l'$ and $l$ are set to 1e-3, and the Adam optimizer is used. The parameters of EnAd-Module are fixed. And 1 NVIDIA A100 80GB GPU is used.

### 4.2 Few-shot Learning for Function Prediction (FuncDesign)

#### 4.2.1 Fold-independent Prediction.
Meta Learning for Fold independent prediction aims to predict unseen enzyme fold classes. Based on $FuncDB_{bi}$, data from 9 folds are used as the training set, leaving one fold as the evaluation. Due to the limited samples in each fold and their mutual independence, we treat each fold as an independent task, leading to the application of meta-learning

| Types | Ours | | | ESM-1F | | |
|---|---|---|---|---|---|---|
| | **M** | **Z** | **F** | **M** | **Z** | **F** |
| *Fold\0* | 0.923 | 0.897 | 0.869 | 0.906 | 0.879 | 0.863 |
| *Fold\1* | 0.915 | 0.907 | 0.864 | 0.872 | 0.841 | 0.813 |
| *Fold\2* | 0.932 | 0.894 | 0.852 | 0.906 | 0.888 | 0.869 |
| *Fold\3* | 0.880 | 0.869 | 0.843 | 0.850 | 0.846 | 0.841 |
| *Fold\4* | 0.925 | 0.897 | 0.869 | 0.907 | 0.872 | 0.847 |
| *Fold\5* | 0.879 | 0.863 | 0.835 | 0.889 | 0.872 | 0.860 |
| *Fold\6* | 0.906 | 0.888 | 0.847 | 0.879 | 0.832 | 0.801 |
| *Fold\7* | 0.880 | 0.869 | 0.852 | 0.860 | 0.850 | 0.813 |
| *Fold\8* | 0.889 | 0.872 | 0.860 | 0.880 | 0.863 | 0.832 |
| *Fold\9* | 0.896 | 0.888 | 0.877 | 0.863 | 0.855 | 0.822 |
| **Avg.** | **0.903** | **0.884** | **0.857** | 0.881 | 0.860 | 0.836 |
| **(Std.)** | (0.020) | (0.015) | (0.013) | (0.020) | (0.018) | (0.024) |

**Table 1: Comparison of enzyme function prediction at the unseen fold level. M: Meta-finetuning; Z: Zero-shot learning; F: General finetuning. Avg.(Std.) denotes the average and standard deviation of results across all folds. Fold\i indicates training on all folds except Fold-i and evaluating solely on Fold-i.**

for task-based learning. 'Fold' here denotes the three-dimensional configuration of secondary structural elements, such as alpha helices and beta sheets, that define a specific protein or protein group. Proteins sharing similar folds usually exhibit substantial structural resemblances, despite variations in their sequences and functions. Classifying folds can provide valuable insights into the evolutionary connections between proteins. As shown in the table 1, Ours(M) involves meta-leaner as a base, followed by meta-finetuning for further refinement. And Ours(Z) directly employs zero-shot learning for trained meta-learner. Ours(M) yields superior results, highlighting the significant improvement brought about by meta-finetuning.

#### 4.2.2 Fold-agnostic Prediction.
Ours(F) serves as a control group, primarily comparing few-shot learning with conventional finetuning methods. It represents **fold-agnostic prediction**, excluding meta-learning learning. We combine positive and negative samples from 9 folds, randomly allocating 80% as the training set and 20% as the evaluation set, treating the remaining 1 fold as an out-of-distribution generalization. For unseen folds, we observed that task-specific learning based on meta-learning is more effective with the average value of Ours(M) at 0.903 exceeding that of Ours(F) at 0.857. Additionally, ESM-IF, acting as a baseline due to its outstanding representation and similar modalities, clearly demonstrates the superiority of our model in terms of generalization.

#### 4.2.3 Enzyme Fold Recognition.
Furthermore, we conduct predictions on more challenging fold types based on $FuncDB_{multi}$, essentially involving a less data-intensive multi-task enzyme function prediction, as shown in Figure 4. This setup aims to predict, within enzymes with mixed folds, the specific fold to which an enzyme belongs. We select ESM-IF and GVP as structure-to-sequence models as baselines, as they also utilize N, $C_\alpha$, and C as standard inputs. These comparisons validate the superior structural representation capabilities by a large margin (Ours: 17.3%P@1,31.7%P@2,41.0%P@3). We employ ESM-2 as a language modality input for comparison to confirm its sequence-only representation capabilities. Although ESM-2 (11.5%P@1,24.5%P@2,36.5%P@3) is a little better than the

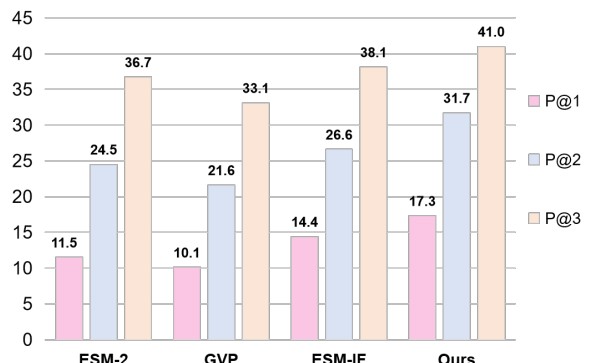

**Figure 4: Comparing the folding prediction abilities of different models. P@k denotes the top-k precision.**

earlier GVP, but is far below the performance of our models, which demonstrates enhanced generalization ability due to the incorporation of structural information.

Overall, while we have undertaken initial explorations in the few-shot setting, the overall accuracy remains relatively low. This is attributed to limitations in the quantity of available data and the diversity of fold types. This compels us to collect data encompassing a broader range of categories.

## 4.3 Non-parametric Zero-shot Learning for Mutation Effects (MutDesign)

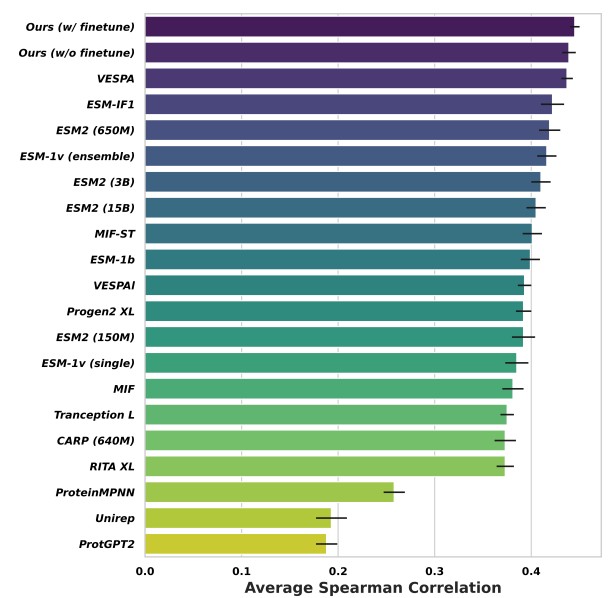

**Figure 5: Spearman's rank correlation between predicted scores and experimental measurements on ProteinGym. Comparison among protein language models and inverse folding models.**

*4.3.1 Mutant Effect Prediction.* To ensure a comprehensive and unbiased comparison, we selected prominent protein language models and inverse folding models as benchmarks, as illustrated in Figure 5

(a). All baseline models employ zero-shot learning for mutation effects evaluation through the $MutDB_{ProtGym}$, which encompasses millions of mutations. Note that our model consists of two configurations: Ours(w/o finetune) directly infers based on UniProt-Net, yielding an average $\rho$ of 43.9%; And Ours(w/ finetune) utilizes the meta-finetuned UniProt-Net, resulting in an average $\rho$ of 44.5%, which consistently achieves the best matching rank, implying that the in-domain finetuning on enzyme data contributes to enhanced overall performance in mutation prediction. In contrast, the optimal protein language model (VESPA)[31] achieved an average $\rho$ of 43.7%, while the optimal inverse folding model (ESM-IF) scored 42.2%. This performance superiority can be attributed to our model's dual advantage, encompassing both contextual and structural transfer learning within the proposed training paradigm.

| | Throughputs | | | | | | | | | | FLOPs | #Params |
|---|---|---|---|---|---|---|---|---|---|---|---|---|
| #Mutant | 500 | 1000 | 1500 | 2000 | 2500 | 3000 | 3500 | 4000 | 4500 | 5000 | | |
| Baseline | 41.0 | 41.5 | 42.7 | 42.2 | 41.2 | 41.5 | 41.9 | 41.3 | 41.1 | 41.3 | 52.7k | 124.82M |
| Ours | 17k | 34k | 51k | 67k | 83k | 101k | 116k | 126k | 131k | 133k | | |

**Table 2: Space and time complexity analyses conducted with an average input protein length 500. Throughputs: samples per second; FLOPs: Gigabyte.**

*4.3.2 Complexity of Decoupled Scoring Mechanism.* The complexity analysis was carried out utilizing a single NVIDIA A100 80GB GPU, as outlined in Table 2. We utilize the widely accepted fitness scoring metric, as referenced in ESM-1v [32], as our baseline. By maintaining a fixed batch size of 1 and an average protein length of 500 amino acids, we vary the number of mutant proteins from 500 to 5000. Space complexity is governed by the number of parameters, where both the baseline and our decoupling method exhibit relatively small parameter counts (124.82M). Regarding time complexity, reflected through Throughputs and FLOPs, both the baseline and our method showcase nearly identical FLOPs (52.7k). As we increment the average number of mutant proteins from 500 to 5000, our decoupling method's advantage further amplifies, although this ascent in advantage slows down due to hardware memory limitations. Our approach requires only one inference step for the wild-type protein to acquire its probability distribution, subsequently facilitating swift calculation of mutant differences between mutant sequences and the inferred wild-type distributions. This significantly expedites the process. In contrast, the baseline necessitates model inference for each mutant protein, resulting in reduced speed. Overall, our model demonstrates markedly enhanced throughput performance, especially noticeable under conditions involving a substantial number of mutations. Compared to the mainstream baseline scoring method, ours boasts a speed advantage ranging from 4k to 30k times faster. This presents a promising avenue for future research, particularly in multi-site/high-order mutation inference, thereby expanding the possibilities for exhaustive exploration.

## 4.4 Parametric Conditional Protein Generation (SeqDesign)

Switching to the NAR mode and coordinating with the lower triangular mechanism, MetaEnzyme is employed for SeqDesign. The amino acid recovery (AAR) metric is adopted.

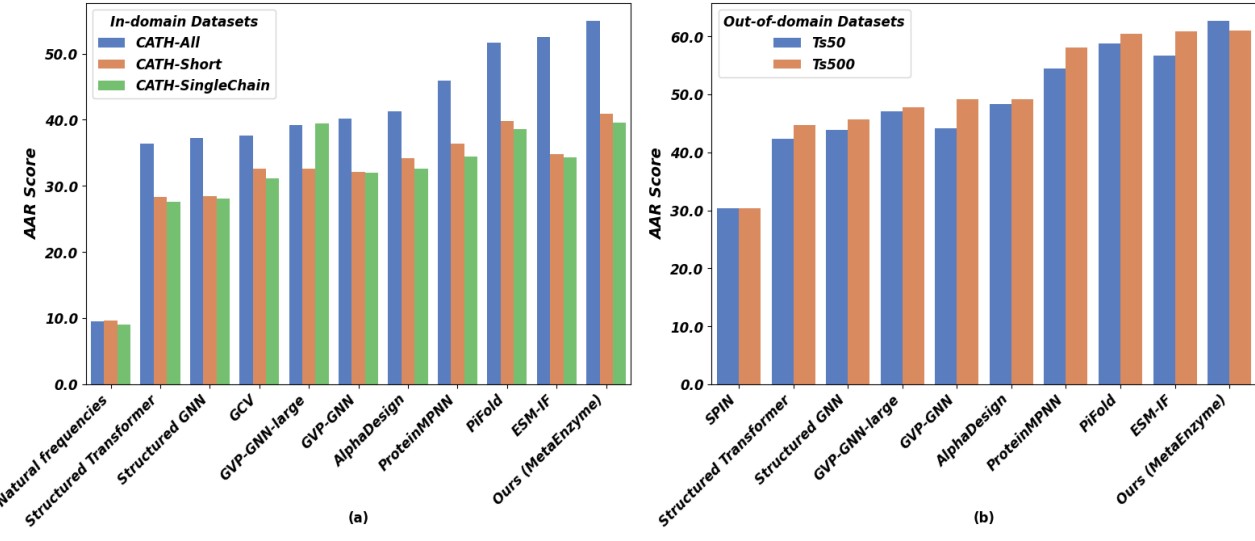

**Figure 6: Comparison on the CATH, Ts50, and Ts500 datasets. The *Short* and *Single-chain* are subsets of CATH test set *ALL*.**

*4.4.1 Internal Validation of General Proteins.* Figure 6 shows SeqDesign comparisons at general proteins, with detailed benchmarks in *Appendix*.

**In-domain Protein Assessment.** In the evaluation of the comprehensive $SeqDesignDB_{ALL}$ ('ALL'), our MetaEnzyme demonstrates superior performances, achieving an impressive AAR score of 54.94%. This surpasses the mainstream ProteinMPNN, PiFold, and ESM-IF consistently by a substantial margin. In alignment with [16], we further assess specific subsets within 'ALL', namely the 'Short' dataset (comprising protein sequences with a length ≤ 100 residues) and the 'Single-chain' dataset (consisting of single-chain proteins cataloged in the Protein Data Bank). Notably, MetaEnzyme exhibits outstanding performance on both 'Single-chain' (39.17%) and 'Short' datasets (40.92%) compared to alternative methods.

**Out-of-domain Generalization.** For a comprehensive comparison of the generalizability to out-of-domain datasets, we present results for the $SeqDesignDB_{Ts50}$ and $SeqDesignDB_{Ts500}$ datasets. Even in these diverse contexts, MetaEnzyme consistently demonstrates improvements, achieving a remarkable AAR score of 62.68% in Ts50 and a breakthrough AAR score of 60.77% in Ts500.

*4.4.2 Generalization on Functional Enzymes.* **Conditional SeqDesign for PET Enzymes.** Polyethylene terephthalate (PET) is a widely used synthetic plastic polymer globally, known for its chemical inertness due to ester bonds and aromatic nuclei. This makes PET resistant to degradation, raising environmental concerns. Enzymatic degradation offers a promising and eco-friendly solution to address the ecological challenge of plastic waste, particularly polyester waste recycling. Despite the potential, our understanding of PET-degrading enzymes is limited. Machine learning-aid techniques [9, 29, 33] might accelerate the discovery of PET hydrolases. To support this effort, we curated the $SeqDesignDB_{Pet}$ dataset for SeqDesign analysis, serving as a valuable resource. Figure 7(PET) illustrates AAR scores for PET hydrolases, showing performance improvements up to 64.35% over baselines (62.56% ESM-IF, 45.40%

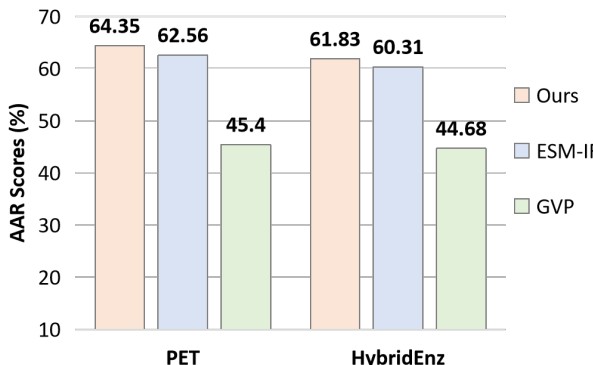

**Figure 7: AAR evaluation of functional enzyme design.**

GVP). This suggests strong generalization capabilities for unseen functional enzymes, paving the way for future analysis and PET-enzyme redesign.

**Conditional SeqDesign for Fold-aware Enzymes.** In the examination of enzyme performance across diverse fold levels within the SeqDesign task, we systematically categorized them into 10 distinct classes utilizing $SeqDesignDB_{Hybrid}$, as presented in Figure 7(HybridEnz). When compared to alternative SeqDesign models, our attained AAR performance of 61.83% consistently outperforms the baseline scores (60.31% and 44.68%). This noteworthy result underscores a persistently elevated overall performance across a spectrum of hybrid fold types, emphasizing the robustness and effectiveness of our SeqDesign model.

## 4.5 Initialization Analysis and Ablation Study

To underscore the significance of pretrained modules, we conducted an ablation study to assess the impact of the pretrained EnAd-Module (PEM) and the pretrained Context-Module (PCM). The results, outlined in Table 3, highlight their substantial contributions to overall improvement. Specifically, when comparing #1 vs. #3

| # | PEM | PCM | Perplexity | | Recovery(%) | |
|---|-----|-----|------|------|------|------|
| | | | Val | Test | Dev | Test |
| 1 | ✗ | ✗ | 7.02 | 6.83 | 36.22 | 37.04 |
| 2 | ✗ | ✓ | 6.13 | 6.40 | 40.11 | 42.21 |
| 3 | ✓ | ✗ | 4.80 | 4.49 | 47.65 | 48.25 |
| 4 | ✓ | ✓ | **4.03** | **3.88** | **53.15** | **54.94** |

Table 3: Ablation study of pretrained modules on CATH validation and test set (PEM: Pretrained EnAd-Module; PCM: Pretrained Context-Module). The best results are bolded.

and #1 vs. #2, it is evident that the pretrained structural features play a pivotal role in boosting performance. This aligns seamlessly with our rationale for incorporating prior structural knowledge. Moreover, given that the PCM module operates downstream of PSM, it is likely to directly benefit from the representation capabilities of PSM. Surprisingly, in the isolated comparison of #1 vs. #2, significant performance is enhanced even when utilizing only the PCM without the initialization parameters of the PEM. This intriguing finding, previously unexplored in SeqDesign studies, suggests that the PCM has acquired substantial knowledge of the protein language, thereby contributing significantly to improved SeqDesign performance. This observation gains further support in the comparison between #3 and #4, solidifying the conclusion that the PCM's acquired knowledge plays a vital role, diminishing the reliance on initialization parameters. The comprehensive analysis presented here highlights the logical and noteworthy advancements brought about by these pretrained modules in protein design.

## 5 IN VITRO WET EXPERIMENTAL VALIDATION

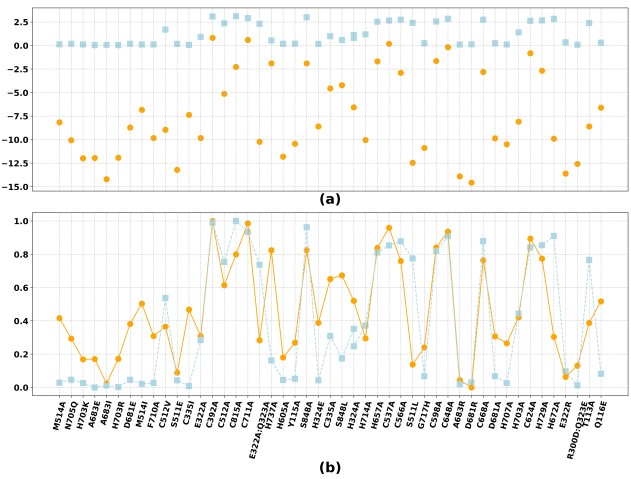

Figure 8: Mutation effect analysis of P-protein. (a) The effects predicted by our model alongside the reference experimental effects. (b) The normalized reference scores and predicted scores offer an intuitive distribution trend. Light blue squares indicate reference values, while orange circles represent predicted values.

To further validate the reliability, we selected a commercially relevant enzyme with potential applications in production, the reversible glycine cleavage system (rGCS) [28, 39, 48], for wet lab experimentation. rGCS, known for efficiently fixing carbon to produce glycine in vitro, currently exhibits relatively low yields in glycine production. Traditional rational enzyme design for rGCS has shown limited improvements in catalytic activity and other attributes. The rGCS, consisting of three proteins (P-protein decarboxylase, T-protein aminomethyltransferase, and H-protein shuttle), can fix two different one-carbon carbon sources to synthesize the two-carbon compound glycine. We focus on P-protein redesign in this work. For details on the structure and mutation preparation processes, please refer to *Appendix*.

Utilizing rational design principles, we strategically chose mutation sites with anticipated substantial influence on mutation effects. Wet lab experiments assessing carbon fixation were conducted for approximately 40 guided mutations, and their effects are illustrated in Figure 8. The wet lab outcomes exhibit close alignment with in silico predictions, demonstrating a notable Spearman's $\rho$ ranking correlation of 70.1%. These results reinforce the practical efficacy of MetaEnzyme, pinpointing crucial sites for subsequent iterations in enhancing the P-protein's carbon-fixing capabilities.

## 6 RELATED WORK

In the dynamic landscape of protein design, AI-driven approaches, exemplified by inverse folding [2, 13, 38, 50], have made significant strides. Tools such as Structured Transformer [16] and GVP-GNN [19] have pioneered conditional protein generation, while recent models like ProteinMPNN [8], PiFold [8], and ESM-IF [15] showcase advancements in sequence recovery. AlphaFold2 [20], RosettaFold [3], OmegaFold [46], helixfold [10], ESMFold [27] stand out as influential structure prediction models. In the niche domain of enzyme engineering [4, 14, 21–25, 40, 43], language and structure models such as DeepSequence [44],ESM-1v [32, 41, 42], Tranception [35], ESM-2 [27] employ few/zero-shot learning for mutation fitness. Our work extends these innovations, leveraging learned structural and linguistic insights, to encompass comprehensive design tasks, addressing challenges in general proteins and functional enzymes. Notably, the field grapples with the need for broader generalization across diverse proteins and presents opportunities for advancing functionality prediction.

## 7 LIMITATIONS AND CONCLUSIONS

This work presents a pioneering MetaEnzyme framework tailored for crucial enzyme tasks, unveiling an architectural breakthrough with far-reaching implications across diverse protein engineering applications. MetaEnzyme underscores the prospect of transitioning from a universal to a unified enzyme design, enabling seamless adaptation across various functionalities through straightforward architectural modifications or lightweight adjustments. Despite the resource-intensive wet lab validation in this study, certain **limitations** are acknowledged: 1) restricted datasets of functional enzymes; 2) the necessity for more robust models to enhance generalization; and 3) a desire for additional wet lab experiments. Overcoming these challenges is crucial for advancing the field, and bolstering the applicability of enzyme design models.

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
