# OpenReview forum: "MetaEnzyme: Meta Pan-Enzyme Learning for Task-Adaptive Redesign"
_acmmm.org/ACMMM/2024/Conference — MM2024 Poster_

### Official Review · Reviewer_hEqg · 2024-05-24

**Rating:** 4
**Confidence:** 3

**Summary:**

## Summary
"MetaEnzyme: Meta Pan-Enzyme Learning for Task-Adaptive Redesign" presents an innovative framework that addresses significant challenges in the field of enzyme design. By leveraging a meta-learning approach, the paper introduces a unified, task-adaptive enzyme design process that integrates cross-modal structure-to-sequence transformation architectures. The framework focuses on three key enzyme design tasks: functional design (FuncDesign), mutation effect prediction (MutDesign), and sequence generation design (SeqDesign). The results demonstrate substantial improvements in enzyme design performance across various tasks, showcasing the adaptability and efficiency of MetaEnzyme.

# Questions:
1. How does MetaEnzyme incorporate structural similarity metrics when designing and evaluating enzymes? Specifically, are metrics such as RMSD or TM-score used to ensure that the designed enzymes retain the necessary structural integrity and functional conformation?
2. Given the importance of the active site configuration for catalytic activity, how does the model ensure that structural modifications do not adversely affect the enzyme's active site geometry?

**Strengths:**

**Unified Framework**: MetaEnzyme effectively consolidates multiple enzyme design tasks into a single, cohesive framework, facilitating better task adaptation and efficiency.

**Comprehensive Evaluation**: The framework is thoroughly evaluated on multiple datasets, including CATH and SeqDesignDB, showing notable improvements in enzyme function prediction, mutation effect prediction, and sequence generation

**Limitations:**

The paper does not provide sufficient details on how structural similarity metrics, such as RMSD (Root Mean Square Deviation) or TM-score, are utilized in the design and evaluation of enzymes. Considering the critical role of enzyme structure in catalytic function, the incorporation of these metrics would be essential for ensuring the designed enzymes retain functional conformations.

The framework heavily relies on pre-trained modules. While these pre-trained components enhance performance, they also introduce a dependency on the availability and quality of pre-trained data, which may not always be accessible or sufficiently comprehensive for all enzyme design tasks.

While the paper provides an overview of the algorithms used, a more detailed explanation of the specific techniques and their implementations would help in understanding the precise mechanisms through which MetaEnzyme achieves its results. This includes a deeper dive into the meta-learning and domain adaptation strategies employed.

**Suitability:**

2

---

### Official Review · Reviewer_8g1U · 2024-05-30

**Rating:** 3
**Confidence:** 3

**Summary:**

Enzyme design is crucial in both industrial production and biology. However, due to the lack of comprehensive benchmarks and the complexity of enzyme design tasks, computational enzyme design remains in its early stages. In this paper, we introduce MetaEnzyme, a staged and unified enzyme design framework, to address these challenges. MetaEnzyme employs a cross-modal structure-to-sequence transformation architecture to obtain robust protein representation, and leverages domain adaptive techniques to generalize specific enzyme design tasks under low-resource conditions. The framework focuses on three fundamental low-resource enzyme redesign tasks: functional design (FuncDesign), mutation design (MutDesign), and sequence generation design (SeqDesign). Through novel unified paradigm and enhanced representation capabilities, MetaEnzyme demonstrates adaptability to diverse enzyme design tasks, yielding outstanding results. Wet lab experiments further validate these findings, reinforcing the efficacy of the redesign process.

**Strengths:**

1.	Innovation: The MetaEnzyme framework has been proposed, which effectively solves the problems of data scarcity and insufficient model generalization in protein design through a cross modal structure to sequence transformation architecture and domain adaptation technology.
2.	Universality: The MetaEnzyme framework is suitable for various enzyme design tasks in low resource environments, including functional design, mutation design, and sequence generation design.
3.	Experimental verification: The effectiveness of the MetaEnzyme framework was verified through wet experiments, further demonstrating the reliability and efficiency of the redesign process.

**Limitations:**

1.	Experimental comparison: The lack of comparison with traditional protein design methods, as well as more standard methods, limits a comprehensive understanding of the advantages of the MetaEnzyme framework in practice.
2.	Quantitative analysis: Lack of deeper quantitative analysis of experimental results, such as error analysis or trend analysis of model performance with data size changes, limits the interpretation and promotion of results.
3.	Terminology explanation: Some terminology explanations are not clear enough, which may cause confusion for readers. It is necessary to supplement some terminology tables to explain the meanings of professional terms.

**Suitability:**

2

---

### Official Review · Reviewer_fHfk · 2024-06-08

**Rating:** 4
**Confidence:** 3

**Summary:**

This paper introduces MetaEnzyme, a new framework for computational enzyme design that addresses the challenges of data scarcity and model generalization in the enzyme design field. MetaEnzyme leverages a cross-modal structure-to-sequence transformation architecture to obtain robust protein representations and employs domain adaptive techniques to generalize enzyme design tasks under low-resource conditions. It focuses on three fundamental low-resource enzyme redesign tasks: functional design (FuncDesign), mutation design (MutDesign), and sequence generation design (SeqDesign). It propose geometry-enhanced module, combined with techniques such as data augmentation and multi-modal fusion to enhance generalization in low-resource scenarios. Results demonstrate MetaEnzyme's adaptability and effectiveness across diverse enzyme design tasks. And wet lab experiments further validates its findings.

**Strengths:**

1. The proposed MetaEnzyme framework can handle different tasks where systematic research has been lacking. It  contributes to the broader field of enzyme engineering.
2. The paper is well-structured, with clear sections and a logical flow of information. Lots of formula expressions make it easy for readers to understand the methods.
3. The proposed mutation effect scoring method is novel and effective, as it offers significant computational efficiency
  especially for sequence datasets with a large number of entries.

**Limitations:**

1. As the datasets are limited, more experiments and analysis with extensive datasets are needed to clarify its effectiveness and generalization.
2. The method relies on pretrained universal protein models, which could potentially limit the model's performance on tasks that differ significantly from the data the pretrained models were exposed to.

**Suitability:**

3

---

### Meta-Review · Area_Chair_4nG9 · 2024-07-07

**Recommendation:** Accept (Poster)
**Confidence:** 4

**Metareview:**

This paper proposes MetaEnzyme for computational enzyme design. It employs a cross-modal structure-to-sequence transformation architecture to establish robust protein representations and utilizes domain adaptive techniques for various enzyme design tasks under low-resource conditions. Specifically, MetaEnzyme focuses on functional design, mutation design, and sequence generation design. Wet lab experiments validate its efficacy, highlighting its potential impact on enzyme redesign.

Initially, the paper got 1 br and 2 ba. The main concerns are about the lacking experiments and its generalization ability. All reviewers have not give the final rating. The AC have read the authors rebuttal, which provide experiments on more datasets, the quantitative analysis, and more method details. The AC find the main idea of the MetaEnzyme interesting, and the experiments are sufficient and supportive.